# Testing Human-Robot Interaction in Virtual Reality: Experience from a Study on Speech Act Classification

**Sara Kaszuba**
kaszuba@diag.uniroma1.it
Sapienza Università di Roma
Rome, Italy

**Sandeep Reddy Sabbella**
sabbella@diag.uniroma1.it
Sapienza Università di Roma
Rome, Italy

**Francesco Leotta**
leotta@diag.uniroma1.it
Sapienza Università di Roma
Rome, Italy

**Pascal Serrarens**
pse@pale.blue
PaleBlue
Stavanger, Norway

**Daniele Nardi**
nardi@diag.uniroma1.it
Sapienza Università di Roma
Rome, Italy

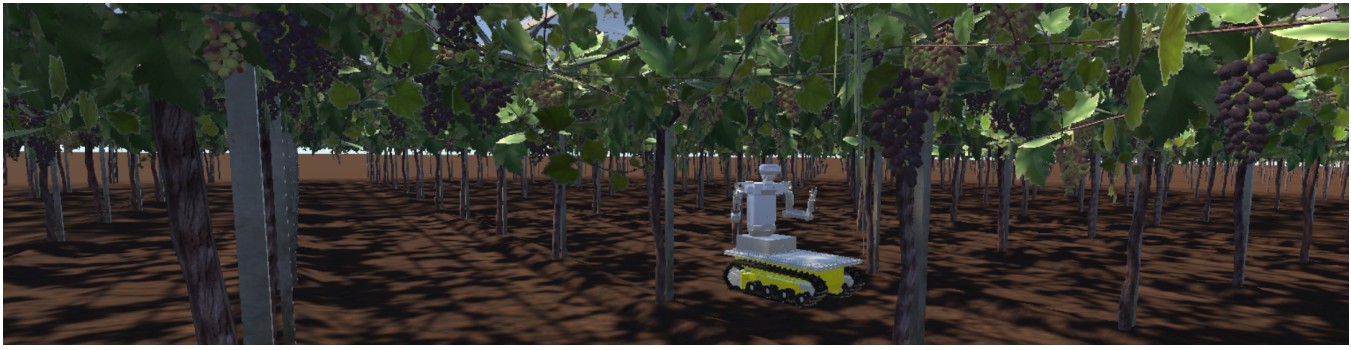

**Figure 1: Virtual Reality environment used to conduct our experiments in both immersive and non-immersive experiences.**

## ABSTRACT

In recent years, an increasing number of Human-Robot Interaction (HRI) approaches have been implemented and evaluated in Virtual Reality (VR), as it allows to speed-up design iterations and makes it safer for the final user to evaluate and master the HRI primitives. However, identifying the most suitable VR experience is not straightforward. In this work, we evaluate how, in a smart agriculture scenario, immersive and non-immersive VR are perceived by users with respect to a speech act understanding task. In particular, we collect opinions and suggestions from the 81 participants involved in both experiments to highlight the strengths and weaknesses of these different experiences.

## CCS CONCEPTS

• **Human-centered computing** → **User studies**; *Usability testing*; **Empirical studies in interaction design**; **Virtual reality**; • **Computing methodologies** → Information extraction; • **Applied computing** → **Agriculture**.

## KEYWORDS

Virtual Reality, Human-Robot Interaction, Smart Agriculture, User Evaluation

**ACM Reference Format:**
Sara Kaszuba, Sandeep Reddy Sabbella, Francesco Leotta, Pascal Serrarens, and Daniele Nardi. 2018. Testing Human-Robot Interaction in Virtual Reality: Experience from a Study on Speech Act Classification. In *Proceedings of VAM-HRI '23*. ACM, New York, NY, USA, 8 pages. https://doi.org/XXXXXXX.XXXXXXX

## 1 INTRODUCTION

The growing interest in novel technologies, such as Virtual Reality (VR), led researchers to change the way in which user studies are conducted in the field of Human-Robot Interaction (HRI). In fact, it has been shown that evaluating preliminary solutions or complete applications in VR could provide multiple benefits in terms of experimental settings, costs, and safety [11]. To this aim, a replica of the real scenario is a reasonable alternative in training people in specific tasks [13, 16, 22], validating potentially harmful actions [6, 14], testing the functionality of the developed approaches [7, 17] and conducting multiple experiments. Hence, the application of VR in all its forms of experience (either immersive or non-immersive) is highly employed in different fields of study, such as medicine, education, industry, aerospace, architecture and history. Essentially, immersive VR allows the user to "be part of the environment" by projecting him/her into an entirely digitally generated world. In such a scenario, the person can usually manipulate virtual objects

and interact in the simulated environment as it happens in a Human-Human Interaction. On the contrary, with non-immersive VR, the user sees the virtual scene through a screen without losing the perception of the real world.

Nevertheless, very few studies cover the use of VR in collaborative robotics for precision agriculture scenarios. However, to the best of our knowledge, none of the available works in such a field describes VR as a reliable solution for both data acquisition and system evaluation. In particular, we are interested in investigating the adoption of this novel technology to carry out HRI-related experimental activities and collaboration between human workers and robots, such as in grape harvesting and pruning, in the CANOPIES project[1]. Hence, to enhance the robot's understanding of the human (and vice-versa) while performing collaborative tasks in the table-grape application scenario, we are considering the interaction from an agent communication perspective by analyzing the message type (speech act) exchanged between humans and robots. To this aim, we want to present two user studies conducted in VR and differing in the type of experience: immersive and non-immersive. In both experiments, the 81 participants involved interacted vocally with the agronomic robot in the table-grape vineyard simulated environment of the CANOPIES project by providing utterances with different content information, such as Information, Command, and Request, receiving as verbal feedback the robot's understanding (as a classification of the sentence provided in input), along with the robot's action in response to the command. To obtain a more natural interaction during the execution of collaborative tasks, we consider speech as the primary communication channel. However, we plan to combine voice with other modalities, such as gesture, light, and sound, to avoid uncertainty in communication and misunderstandings. Hence, we will develop a multimodal modular system to facilitate the integration and/or substitution of specific components. The identification of the most suitable classes of communication acts, along with the best interaction modalities to ensure collaboration between humans and robots in table-grape vineyards, is covered in our previous work [8].

In this article, we aim to compare a user study conducted in an Immersive Virtual Environment (IVE) and an experiment performed in a Non-Immersive Virtual Environment (NIVE), so to discuss participants' feelings and identify the most suitable type of experience that could be adopted for the upcoming experiments in the CANOPIES project. In both studies performed in a virtual reality environment created in the project context, the developed speech act classification system was only used as a tool to evaluate and compare the two experiences. Hence, a general view of the speech act classification model, such as the training data and the pipeline, will be presented to the reader without focusing on its effectiveness. To this aim, the research questions we are addressing in this article are the followings:

- Q1: Is the IVE experience preferable to conduct user studies in the table-grape scenario compared to NIVE?
- Q2: How the experience in NIVE could influence users' feelings of the immersive world compared to people participating only in IVE?

The paper is organized as follows: Section 2 provides an overview of previous works employing IVE and NIVE to conduct user studies. Section 3 presents the preliminary system developed for the robot's speech act understanding and used as a tool to compare the two experiments in the immersive and non-immersive table-grape application. Then, Section 4 describes in detail the two different experiences by focusing on the participants involved, the general setup, and the duration of the study. Section 5 highlights the results obtained from the users' experience questionnaire concerning the two types of settings. Then, we conclude the article with a discussion about the emerging strengths and weaknesses of both IVE and NIVE to identify the most suitable experience that can be adopted for future user studies in our application scenario.

## 2 RELATED WORKS

Collaborative robotics in precision agriculture is a recently explored area, and researchers have only begun studying HRI in such scenarios. To this aim, the growing interest in immersive and non-immersive VR in evaluating HRI studies in other disciplines led us to consider adopting VR in both its forms to conduct experiments in collaborative robotics for agriculture. However, to the best of our knowledge, there are no available works investigating IVE and NIVE in such a scenario.

In the educational context, several recent papers highlight the results obtained from a comparison between the experience in NIVE and IVE. The evaluation of immersive VR with respect to non-immersive applications, in terms of learning effects on middle school students with ASD, is presented by Carreon et al. in [2]. In such work, the authors, through a user study, showed that the students achieved notable learning improvements in both experiences, screen-based VR and head-mounted display VR. Interestingly, a significant increase in learning gain, user enjoyment and concentration emerged from the immersive experience described in [12], where Mahmoud et al. conducted an experimental analysis by comparing the impact of immersive and non-immersive systems in learning. The effectiveness of immersive VR in teaching medical students practical skills and clinical interventions is demonstrated by Omlor et al. in [15]. In this article, the authors compare students' learning experiences through VR-viewer and non-immersive screens. Similarly, the potential of IVE in teaching crystal lattices was presented in [20]. Vergara-Rodríguez et al. highlight how the level of immersion could influence relevant design aspects, in particular usability, ease of use, motivation and interactivity. At the same time, Renganayagalu et al. compared immersive and non-immersive experiences in maritime education to train and improve seafarers' skills in [9]. In such a study, higher motivation and preference have been demonstrated towards immersive training simulator engines than non-immersive ones. Stronger memory performance associated with users participating first in the immersive experience emerged from [19], where Ventura et al. conduct a user study with a few people performing a task in IVE, then in NIVE; while, another group began with the task in the non-immersive scenario, moving then to the immersive experience.

Most of these works demonstrate that adopting immersive VR for educational purposes to evaluate proposed solutions is particularly effective. Introducing such novel technology could improve task

[1]https://canopies.inf.uniroma3.it/

performance and increase user satisfaction and the learning process. However, as mentioned at the beginning, HRI in collaborative robotics for precision agriculture represents a very recent research field that is slowly expanding both in the researchers' interest in such applications and the usage of novel technologies, such as VR, AR, MR, and XR. To this aim, we believe that more investigations should be conducted in such a field since it is a promising research area that could significantly impact efficiency, safety, and sustainability.

## 3 SYSTEM IMPLEMENTATION

The growing interest in developing systems and applications for the agricultural field led many researchers to create and share various data collections among the research community. However, regarding the vineyard scenario, most of the dataset released consist of crop and/or fruit images that are required to train computer vision algorithms and perform perception tasks, such as fruit detection, identification and segmentation, disease understanding, and colour recognition. To the best of our knowledge, there is no spoken or textual available dataset to conduct studies and analysis on HRI in table-grape vineyards. Hence, the lack of such data and our goal of examining the interaction between humans and robots from the type of information exchanged during the conversation (speech act), prompted us to acquire an Italian dataset and employ it in training the speech act understanding network. The data acquisition process is described in Section 3.1, while the system is presented in Section 3.2.

### 3.1 Data Collection

The necessity of a speech dataset to support HRI for collaboration in outdoor application scenarios, specifically in table-grape vineyards, led us to consider the idea of acquiring vocal utterances about operations in the vineyard. Since our final goal is to develop an efficient and ready-to-use system to be evaluated and employed in Agrimessina, a table-grape producer in southern Italy, we opted to collect an Italian dataset by involving Italian native speakers in the data acquisition process. Users interested in the experiment were asked to sign a consent document to record their voices.

Hence, based on our preliminary study conducted in [8], we initially collected this data separately with three different scenarios, one for each category under investigation: Information (22 questions), Command (20 questions), Request (20 questions). The customized questionnaires were created through the Jotform[2] platform to reach, with our forms, as many participants as possible. Jotform was chosen for its practicality in utterance recording, facilitating the procedure of speech acquisition from the user perspective and, consequently, speeding up the entire data collection process. In all three forms, we described the general context and some technical terms that users could exploit in generating the vocal responses. Additionally, to avoid uncertainty or misunderstanding in the description of the scenario, a representative image followed each question. Such pictures (some captured from the simulation environment, others in the real field, and a few taken from the web) illustrate tasks, activities, actions, and vineyard elements such as grapes, leaves, branches, and the pergola system.

---

[2]https://eu.jotform.com/

*3.1.1 Participants.* At the end of the acquisition process, we reached around 40 participants with each form (most of the users provided vocal utterances for all three forms). However, we noticed that people who participated in the data acquisition process mainly were males between 18 and 30 years old.

Regarding the level of experience with robots and expertise in the vineyard, we detected quite balanced robotic skills among the participants, with half of them having good proficiency or interacting several times with a robot. Nevertheless, looking at the expertise in the vineyards, we noticed that most people had little knowledge about the activities conducted in such an environment. However, such results are reasonable since we mostly spread the forms with Bachelor, Master, and PhD students in Engineering in Computer Science at our university. From this point of view, unfortunately, recruiting a sufficient number of vineyard operators was impossible. In addition, we imagine such kind of system to be more of interest to the next generation of digital farmers, of which young students may represent a reasonable proxy.

*3.1.2 Utterances.* The data acquisition process of the Information, Command, and Request speech act categories lasted around two months. At the end of this period, we collected around 1,800 vocal utterances with their corresponding textual transcriptions. All the details concerning the data analysis are provided in Table 1, such as the number of sentences before and after removing duplicated utterances and the final number of statements obtained after re-arranging them accordingly to the category of the content information that they represent.

From Table 1, it is clearly noticeable that the categories involved in the sentence adjustments are *Command* and *Request*. Such an outcome is reasonable, as the difference between the two aforementioned classes could be very subtle from the user's perspective. We listed under the 'Command' speech act, all the sentences provided by the human that the robot must execute in the short term, such as "Turn left", "Harvest the ripe grape bunches", or "Remove all the dry branches". At the same time, utterances asked by the person that do not require an immediate interruption of the robot's activity have been included in the 'Request' class. For instance, sentences belonging to this category are "Can you tell me your battery level?", "Could you help me in harvesting these grapes?", "How many damaged grapes have you identified?".

### 3.2 Speech Act Understanding

The necessity of a modular and robust system that could exploit the speech act classification to simplify the robot's comprehension of the specific information content led us to develop a preliminary ROS framework for spoken interaction between humans and robots. Several existing works that identify speech as the primary communication modality, based on old ROS versions, were used in developing our system, such as [3, 5, 10]. In particular, we implemented our preliminary framework version on ROS Noetic, on a computer running Ubuntu 20.04 LTS. The proposed pipeline used for both immersive and non-immersive experiments is presented in Figure 2. The first module is the *Speech-to-Text*, which is responsible for acquiring human spoken utterances and converting them into their textual representation through Vosk [1], an offline open-source speech recognition toolkit. Vosk was chosen for its higher

**Table 1: Analysis of the acquired data**

| Speech Act | Sentences (with duplicates) | Sentences (without duplicates) | Sentences (after category adjustments) |
|---|---|---|---|
| Information | 900 | 804 | 804 |
| Command | 728 | 517 | 665 |
| Request | 719 | 534 | 342 |

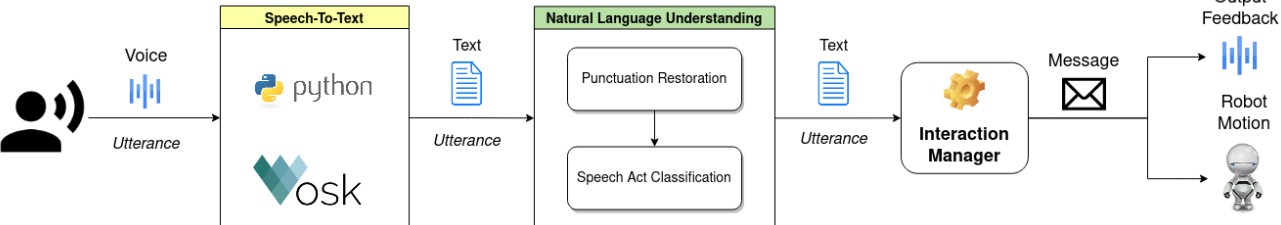

**Figure 2: Developed ROS speech pipeline.**

accuracy in utterance transcriptions with the smallest models and for supporting different languages. Since our final goal is to develop a robust and reliable system with whom we expect people to interact spontaneously, we believe that also allowing communication in different languages (for foreign human workers) could improve communication with the robot.

Subsequently, the textual transcription of the utterance is processed by the *Natural Language Understanding* module. Such a component introduces punctuation in the sentences using the Bert restore punctuation model by Hugging Face[3], to simplify the process of speech act classification. For instance, a question mark at the end of a statement has a higher probability of associating the sentence with the 'Request' class rather than the 'Command' or 'Information' categories. We trained the speech act classification model on the dataset acquired through Jotform by adopting the AutoGluon library [4] for Natural Language Processing (NLP) tasks. We opted for such an automated machine learning library since it could select the best training parameters by optimizing them with respect to the final task (in our case, a classification problem). However, another relevant advantage of this library is associated with the existence of a specific predictor for supporting multimodality, which could help us improve our final system.

Finally, based on the sentence prediction class obtained through AutoGluon, the *Interaction Manager* performs some checks to verify if any of the words of the sentence appears in the system's knowledge and generates a message that is sent to the speakers, so to provide vocal feedback of the robot's speech act understanding ('Information', 'Command', 'Request' or a combination of different categories in case of subordinate sentences). When the robot recognizes the utterance belonging to the 'Command' class, a motion message is sent to the motors of the virtual robot, attaining the corresponding movement in the simulated environment (both immersive and non-immersive). Nevertheless, even if a different category is recognized, the robot provides feedback to the user by

moving its head and looking around to give a feeling of comprehension.

## 4 STUDY DESIGN

Students from our university were mostly involved in the experiments to evaluate immersive and non-immersive experiences in VR. In particular, we decided to include young adults in the studies because we believed they could provide us with more accurate feedback about adopting this novel technology to test HRI approaches in the CANOPIES project. Since the user studies were conducted for approximately one month in a laboratory at our department, we expected a high student engagement.

Before starting the experiment, participants were requested to sign a consent document to record their voices while interacting with the virtual robot. Subsequently, a brief description of CANOPIES and the goal of such experimental activities were provided. In presenting the project, we mainly focused on explaining the agronomic robot's ability, since users would have interacted with it in the virtual environment, and on illustrating the different table-grape varieties, such as "white pizzutello", "black pizzutello", and "black magic" to enrich the communication. Indeed, one drawback of recruiting participants with a pure engineering background was related to potential limitations in their expression due to the lack of knowledge of the specific terminology of the table-grape scenario and activities that can be carried out in a vineyard. During the interaction, we observed the user's behaviour while changing the task. At the end of the experience, each participant was asked to complete a questionnaire by assigning a score from a 5-point Likert Scale, ranging from 1 ("Strongly Disagree") to 5 ("Absolutely Agree") to each of the 23 statements. We based our questionnaire on the user study evaluation on IVEs presented in [18]. However, to adopt a similar questionnaire for both NIVE and IVE (for comparison), we reviewed some utterances and adapted a few to the goal of our experimental activities.

---

[3]https://huggingface.co/felflare/bert-restore-punctuation

## 4.1 Non-Immersive Experience

The experiments in the NIVE were conducted for two weeks, allowing most Master's and PhD students to participate in the user study. Nevertheless the one-hour duration of the experience in NIVE, we reached a good number of participants (40 students). However, we noticed a higher presence of males (32) than females (8) in our evaluation. Moreover, we observed that 75% of the users had already experienced VR applications before this study, and 17.5% of them had negative feelings after using such systems in the past. In analyzing the expertise in the vineyard and with robots (in non-immersive and real scenarios), we discovered that 27.5% of people had experience in a vineyard, and 40% worked with robots in the NIVE. At the same time, 52.5% operated real robots multiple times.

For the experiments in the NIVE, we required a computer with high computational power to run the Unity Linux executable of the CANOPIES non-immersive virtual simulator (generated by Pale-Blue, our Norwegian partner) and the developed speech system. To this aim, we employed an Alienware Aurora Ryzen Edition R14 computer with 64GB of RAM and an AMD Ryzen 9 5950X 16-Core processor. The study was conducted on a machine running Ubuntu 20.04. In this experiment, to reduce the environmental noise and allow the recording of the vocal utterance in separate files by the system, we opted for a headset with a switcher to activate/deactivate the microphone. Moreover, in the NIVE, the participant could move the avatar (representing himself/herself) through the keyboard and change the horizontal camera view either with the keyboard or the mouse. Hence, the user could supervise the robot while performing a motion autonomously, chosen based on the recognized speech act category and the match with specific words in its knowledge base.

## 4.2 Immersive Experience

Approximately one week was required to perform studies in the IVE of the CANOPIES project. As for the NIVE, also for this experience, most of the involved students were enrolled in a Master's or PhD in Engineering in Computer Science. With such a study, we were able to attract 41 students, identifying two groups: people that also participated in the NIVE experience (25) and users that were only included in this experiment (16). Similarly, as in the study in the non-immersive scenario, we observed that most of the participants involved were males (33), with only a few females (8). Then, 61% already used immersive VR applications in the past, and 19.5% of them had a negative experience with such technology. Regarding the expertise in the vineyard, 43.9% of the users had sufficient knowledge of the activities in such field. However, 48,8% of the participants had experience with real robots, while only 17,1% with androids in immersive environments.

To conduct our studies in the IVE, we employed two powerful machines: an Alienware Aurora Ryzen R14 computer with 64GB of RAM and an AMD Ryzen 9 5950X 16-Core processor running Ubuntu 20.04 (also adopted in the NIVE experience) and an Alienware x17 R2 with 32GB of RAM, a CPU of 12th generation i9 and Nvidia RTX 3080ti GPU notebook running Windows 11 Home. The Alienware Aurora Ryzen ran the developed speech system. At the same time, the Alienware x17, connected to the Oculus Quest 2 headset (through Oculus Link cable), allowed us to analyze what the

user was observing during the interaction. The maximum exposure time in the IVE was set to half an hour to reduce the probability of experiencing any discomfort. In such a scenario, the communication device adopted was the Jabra microphone (that participants held in their hands during the interaction) to avoid ambient noise and allow the recording of the vocal utterances in separate files by pressing the microphone button to activate/interrupt the acquisition process. Since users could change the avatar's position (representing themselves) in the NIVE, we decided to let people also move in the IVE. To this aim, we identified a 2m x 2m squared space in our laboratory to permit users to experience motion in the immersive vineyard while analyzing the robot's responses and activities. However, unlike the experiment conducted in the NIVE, the robot motion in this user study was not directly controlled by the speech system running in ROS since, by the time we conducted the tests, there was not yet a stable connection between ROS and Unity. For this reason, we decided to faithfully reproduce the robot's movements through the Oculus Quest 2 controllers by relying on the robot's comprehension of the utterance pronounced by the person.

## 5 RESULTS

The outcome of the questionnaires from user studies in the NIVE and the IVE are presented and discussed in this Section. Precisely, the results of the first 23 questions are reviewed based on their measurement categories: Immersion, Presence, Engagement, Flow, Emotion, Skill, Judgement, Experience Consequence, and Technology Adoption. For both studies, we considered a subset of the questions (with associated measurement classes) presented in [18], and we adapted some of them to our application scenario.

Therefore, users' impressions are compared by identifying two macro-groups: people involved in the NIVE (40 subjects) and participants that took part in the IVE (40 subjects). Moreover, users experiencing the immersive scenario can be distinguished into two sub-groups: people participating only in the fully-immersive environment (16 subjects) and a set of users that first experienced the NIVE, then the IVE (25 subjects). Such a choice was driven by our interest in investigating how the experience in the non-immersive scenario could influence users' feelings of the immersive world compared to people participating only in the immersive environment. Hence, the three groups analyzed in the measurements are summarized below:

- people participating only in the first experiment experiencing the NIVE
- people participating only in the second experiment experiencing the IVE
- people participating in the second experiment in the IVE, but they were first involved in the NIVE study

To the sake of completeness, all the questions with the corresponding average score, emerging respectively from each of the aforementioned groups of participants, are available in Table 2, with the highest value for each statement emphasized in bold. At the same time, a graphical representation is provided in Figure 3.

However, by analyzing the 81 questionnaires received, we noticed that considering the Engagement category, most participants found the virtual world's visual aspects very helpful for interacting

**Table 2: English translation of the proposed user experience questionnaire with average scores from the three groups of participants.**

| Statement | Measurement Category | NIVE Experience | Only IVE Experience | IVE with previous NIVE Experience |
|---|---|---|---|---|
| 1. The visual aspects of the virtual environment helped me in the interaction. | Engagement | 4.18 | 4.25 | **4.32** |
| 2. I felt involved in the virtual environment experience. | Engagement | 4.05 | **4.69** | 4.32 |
| 3. I could actively survey what was happening in the virtual environment. | Presence | 4.48 | 3.94 | **4.56** |
| 4. I was able to examine objects closely. | Presence | 3.68 | 4.13 | **4.60** |
| 5. I could examine objects from multiple viewpoints. | Presence | 4.25 | 4.31 | **4.40** |
| 6. I felt proficient in moving and interacting with the robot. | Presence | **4.10** | 3.88 | 3.96 |
| 7. I could concentrate on the task rather than on the devices. | Presence | **4.28** | 4.00 | **4.28** |
| 8. I was so involved in the virtual environment that I was unaware of things happening around me in the real world. | Immersion | 2.73 | 3.31 | **3.40** |
| 9. I was so involved in the virtual environment that I thought I was in the scene. | Immersion | 2.50 | 3.06 | **3.68** |
| 10. I was so involved in the virtual environment that I lost track of time. | Immersion | 2.83 | 3.19 | **3.48** |
| 11. I knew what to say and/or do in each scenario. | Flow | 3.50 | 3.38 | **3.92** |
| 12. I was not worried about people's judgment during the interaction. | Flow | **4.50** | 4.25 | 4.48 |
| 13. Personally, I would say the virtual environment is a valid solution to test the robot's comprehension. | Judgement | 4.48 | **4.56** | 4.36 |
| 14. Personally, I would say the experience in the virtual environment is exciting. | Judgement | 3.65 | **4.19** | 3.84 |
| 15. I felt confident using the keyboard/Oculus Quest 2 and interacting through the headset/microphone (*depending on the type of experience*). | Skill | **4.28** | 4.00 | 3.92 |
| 16. If I use the same virtual environment again, the interaction would be faster and more spontaneous. | Technology Adoption | **4.13** | 4.06 | 4.04 |
| 17. Using devices (headset, keyboard/Oculus Quest 2, microphone) to interact in the simulated environment is simple and practical (*depending on the type of experience*). | Technology Adoption | **4.33** | 4.19 | 4.00 |
| 18. I would like to interact more often with virtual systems similar to this experience. | Technology Adoption | 3.65 | **3.94** | 3.92 |
| 19. I suffered from fatigue, headache, eyestrain, vertigo or nausea during my interaction with the virtual environment. | Experience Consequence | 1.05 | **1.75** | 1.60 |
| 20. I enjoyed the experience so much that I felt energized at the end of the experience. | Emotion | 3.40 | **3.56** | 3.40 |
| 21. During the interaction in the virtual environment, I felt anxious. | Emotion | 1.48 | **1.81** | 1.44 |
| 22. I enjoyed dealing with interaction devices. | Emotion | 3.95 | **4.50** | 4.20 |
| 23. I felt natural interacting vocally in the virtual environment. | Emotion | **4.35** | 3.56 | 3.92 |

with the robot, increasing their feeling of being engaged in the experience. Indeed, we observed an average value of 4 assigned to questions in the Engagement category from all groups. Precisely, the lowest average number (4.11/5) was provided by the group experiencing the NIVE, while the highest one was encountered in people participating only in the immersive experience (4.47/5).

Instead, while evaluating the level of Presence, we considered several aspects, such as the users' ability to actively supervise the robot, examine objects of the environment, such as grapes, leaves, and branches from different distances and perspectives, freely move and interact with the android and concentrate on the task rather than on the devices. Comparing the responses received from the three groups, we discovered that participants involved first in the non-immersive scenario and then in the immersive one appear to have a very high sense of presence in the IVE, equal to 4.36/5. On the

contrary, people participating only in the immersive evaluation had a weaker sense of presence (4.05/5) compared to those partaking in the NIVE experience (4.16/5).

The third measurement category is the level of Immersion, related to the high engagement in the environment that led participants to lose track of time and not be aware of the circumstances happening in the real environment, giving them the feeling of being effectively part of the virtual world. The results obtained from the three groups pointed out that Immersion was definitely encountered in the immersive scenarios rather than in the non-immersive ones (2.68/5). In particular, the highest average value belonged to the group that previously participated in the NIVE experience (3.52/5).

In the Flow measure, we evaluated if users' experience was fluent, knowing what to say and how to behave without worrying

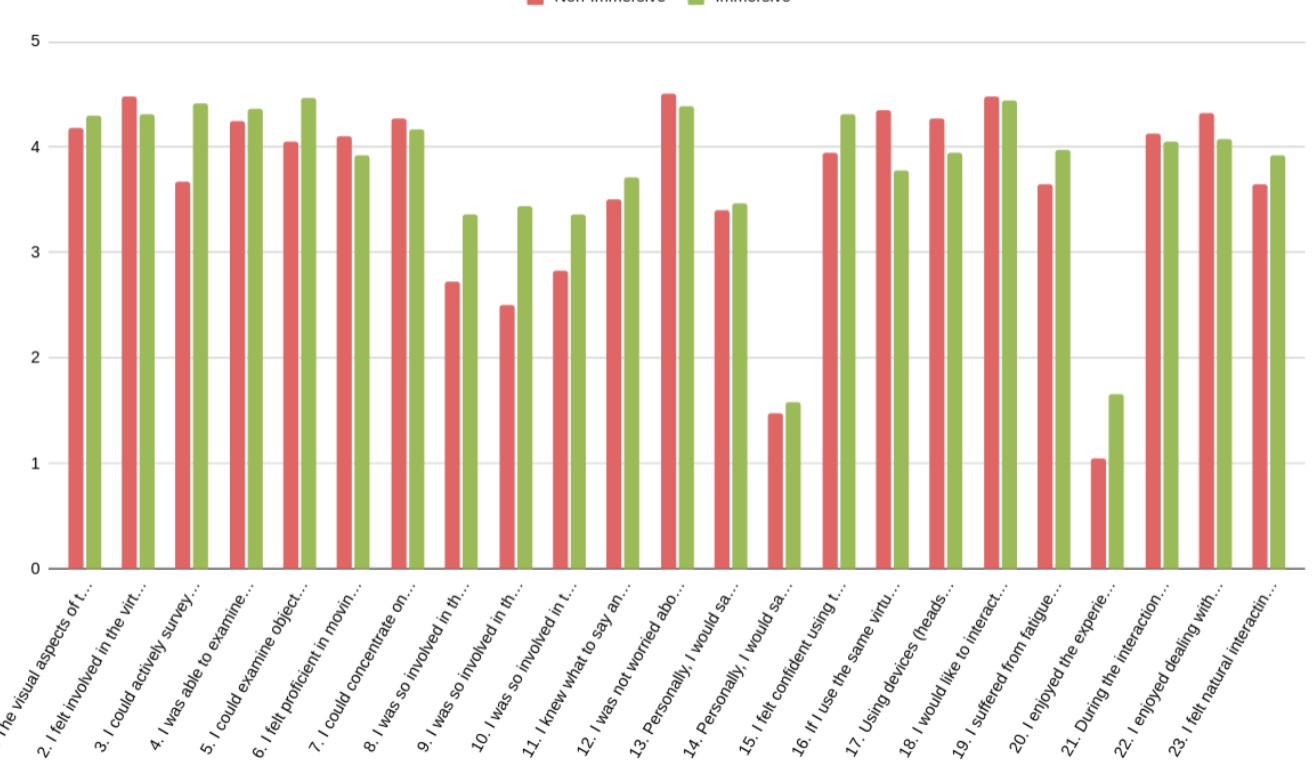

**Figure 3: Graphic representation of the questionnaire responses.**

about external judgments while interacting with the virtual robot. From the responses received, we observed that the highest average value (4.2/5) is assigned to people who participated in the IVE and previously enjoyed the NIVE. In contrast, the lowest average value was encountered in the users that experienced only the immersive scenario (3.81/5).

Through the Judgement class, we aim to obtain feedback concerning adopting the virtual environment as a valid solution to test the robot's comprehension and if the experience was exciting. In this analysis, we noticed that people participating only in the experience in the IVE reached the highest average value (4.38/5). At the same time, in this measurement, the NIVE achieved the lowest average score (4.06/5).

The Skill measure allowed us to identify the users' confidence level with the interaction devices, such as the keyboard and the headset for the NIVE and Oculus Quest 2 with the Jabra microphone for the IVE. The questionnaires highlighted that people participating in the experience in the NIVE felt more confident with the setup (4.28/5) than users that took part in the IVE but were previously involved in the tests in the non-immersive scenario (3.92/5).

The Technology Adoption estimates how much participants are geared and interested in technology. Therefore, we evaluated if users believed that interacting again with the system could improve the speed and communication, if the interaction devices were practical and easy to use and if they would like to interact often in

NIVEs or IVEs similarly as they did in the experience they evaluated. From our analysis, participants involved only in the IVE provided the highest scores for Technology Adoption (4.06/5), followed by the group that examined the NIVE (4.03/5).

The experience Consequence category permits identifying if any user experienced a negative feeling during the interaction, such as fatigue, headache, eyestrain, vertigo, or nausea. From the questionnaires, we expected the highest values to be associated with the IVE experience. In particular, users involved only in the immersive scenario achieved the highest average value (1.75/5), followed by the group that participated in the IVE and previously also in NIVE (1.6/5), while the lowest score belonged to people experiencing the non-immersive scenario (1.05/5).

Finally, we considered the Emotion category to measure positive and negative feelings derived from each experience. Hence, as positive impressions, we evaluated the level of energy people felt after the experiment, the likeliness towards the interaction devices, and the naturalness of spoken communication. At the same time, as negative aspects, we examined the level of anxiety users had during the study. Therefore, we separated the questions belonging to the Emotion category to provide the results associated with each of the aforementioned impressions. Regarding the positive emotions, the scores of the three groups appear very similar, ranging from 3.84/5 (obtained from people experiencing the IVE, but previously were also involved in the non-immersive scenario) to 3.9/5 (achieved by

the participants that took part in the NIVE). However, we noticed that the lowest level of anxiety was found in people who had experienced the IVE and previously participated in the experiment with the NIVE (1.44/5). At the same time, the highest score belonged to the group of users only involved in the IVE (1.81/5).

To show the statistical significance of the results reported in Figure 3, we have applied a two-tailed unpaired T-Test using $p < 0.05$ as a threshold. In particular, a statistically different mean has been identified for questions 3, 5, 8, 9, 16, and 20, whereas questions 10 and 15 assumed borderline values ($p \approx 0.055$). Observed differences in mean for questions 17 ($p \approx 0.06$), 22 ($p \approx 0.07$) and 23 ($p \approx 0.13$) are not statistically significant.

## 6 DISCUSSION

Identifying the most appropriate type of VR experience to conduct specific user studies is not straightforward. Indeed, by analyzing the results of our questionnaires, we noticed that some measure categories (6 of 9) obtained the highest scores from IVE participants: Engagement, Presence, Immersion, Flow, Judgment, and Technology Adoption. To this aim, most users who had previously interacted in the NIVE highlighted their preference for the immersive experience. However, issues related to the prolonged use of the Oculus Quest 2 headset and the reduced motion space were also pointed out in one of the open questions asking for the negative aspects of the experience (if any). Hence, looking at the results of our studies, for the upcoming experiments concerning the CANOPIES project, we aim at adopting immersive user studies mainly for acquiring data and conducting tests to evaluate HRI approaches that do not require people to interact with the digital world more than half an hour and to move physically, so to avoid any form of discomfort. Hence, we would like to prevent immersive experiences when a person's motion is entailed for evaluating proposed solutions, but no sufficient space is accessible to conduct such an experiment. Nevertheless, teleportation could also be employed to allow human motion in limited spaces, but researchers demonstrated that such an unnatural method of place changing could stimulate users' motion sickness [21]. To this aim, we believe that non-immersive scenarios can always be employed in investigating longer HRI studies and solutions requiring humans and robots to evaluate task performances in different areas of the virtual environment.

## ACKNOWLEDGMENTS

This work has been partly supported by the H2020 EU project CANOPIES - A Collaborative Paradigm for Human Workers and Multi-Robot Teams in Precision Agriculture Systems, Grant Agreement 101016906.

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
