# OpenReview forum: "Testing Human-Robot Interaction in Virtual Reality: Experience from a Study on Speech Act Classification"
_humanrobotinteraction.org/HRI/2023/Workshop/VAM-HRI — VAM-HRI 2023 Oral_

### Official Review · Program_Chairs · 2023-02-25
**Accept**

**Rating:** 6
**Confidence:** 5

**Review:**

Review 1:

This work presents a system for interacting in a virtual agricultural environment. A user study compares the immersive virtual environment (IVE) to a non-immersive virtual environment (NIVE). The work also attempts to classify human speech acts to be interpretable by virtual robots.

It appears as though this paper attempts to address multiple issues: training in VR, categorizing speech acts, and assessing the usefulness of an IVE compared to an NIVE. These three different areas need to be more clearly articulated and linked in order for the paper to make the impact that is desired. I recommend revising the introduction to include exactly what research questions are being addressed in this work, and then adjusting the remainder of the paper accordingly to include hypotheses and then key takeaways. Other minor suggestions follow below.

In section 3, it would help to provide an example of each of the categories of utterances, so that the reader has some understanding of what these might be.

Figure 2 with the pipeline is very helpful. It would also be nice to have another figure that visually shows the architecture of the hardware.

Section 5 – It looks like some of the participants received a within-subjects treatment while others were between-subjects. Can you explain this choice?

--------
Review 2:

In this paper, the authors compare immersive vs. non-immersive virtual reality environments for an HRI task in an agricultural setting. They compare human participant ratings among several dimensions and show that the IVE condition rated higher in most categories. I agree with reviewer 1 that the motivation for using speech acts is unclear and should be discussed more thoroughly at the beginning. I would also like to see a short discussion of the effectiveness of the speech act classification model. Additionally, I would like to see discussion of potential drawbacks of recruiting participants from graduate level engineers.

Overall, I think the concept would be of interest to the VAM-HRI community.

---

### Decision · Program_Chairs · 2023-03-02

Accept (Oral)